# AIHWKIT-Lightning: A Scalable HW-Aware Training Toolkit for Analog In-Memory Computing

**Julian Büchel**[1,2]
jub@zurich.ibm.com

**William Simon**[1]
william.simon1@ibm.com

**Corey Lammie**[1]
corey.lammie@ibm.com

**Giovanni Acampa**[1,2]
giovanni.acampa@ibm.com

**Kaoutar El Maghraoui**[3]
kelmaghr@us.ibm.com

**Manuel Le Gallo**[1]
anu@zurich.ibm.com

**Abu Sebastian**[1]
ase@zurich.ibm.com

[1]IBM Research – Zurich, [2]ETH Zurich, [3]IBM Thomas J. Watson Research Center

## Abstract

We introduce AIHWKIT-Lightning, a new toolkit designed for efficient and scalable hardware-aware training of large neural networks deployed on Analog In-Memory Computing (AIMC)-based hardware. The toolkit prioritizes speed and ease of use, addressing the limitations of existing frameworks in training Large Language Models (LLMs) with billions of parameters. AIHWKIT-Lightning leverages dedicated GPU kernels and a streamlined implementation, achieving up to $3.7\times$ faster training at lower memory consumption compared to state-of-the-art toolkits. Benefiting from the increased scalability, we demonstrate near-iso-accuracy on the GLUE benchmark using a RoBERTa model trained on 11B tokens. The toolkit is publicly available at `https://github.com/IBM/aihwkit-lightning`.

## 1 Introduction

An emerging non-von Neumann computing paradigm that has gained traction in recent years is in-memory computing utilizing Non Volatile Memory (NVM) devices [1–4], often referred to as Analog In-Memory Computing (AIMC). AIMC-based accelerators exploit the inherent properties of NVM devices and the principles of circuit laws to execute Matrix-Vector Multiplication (MVM) operations directly within the memory arrays where the weights are stored. This approach eliminates the need for loading weights from memory and requires only the movement and buffering of activations. As a result, AIMC-based heterogeneous computing architectures are set to provide $10\times$ to $140\times$ higher energy efficiency [5], at competitive throughput, compared to modern GPUs.

Doing computation in the analog domain has the downside of being imprecise due to various sources of noise such as drift [6] and programming noise [7]. For neural networks to retain high accuracy when deployed on AIMC-based hardware, networks must be trained in a Hardware-Aware (HWA) manner to enhance their robustness. HWA training is a commonly used method [8, 9] to fine-tune pre-trained models while applying various non-idealities to the model during forward propagation. Many different types of networks trained for a variety of tasks using HWA training have been shown to have little to no drop when deployed on real AIMC hardware [10–12].

38th Second Workshop on Machine Learning with New Compute Paradigms at NeurIPS 2024(MLNCP 2024).

Over the past few years, the number of parameters in neural networks have exploded. With every released model almost certainly surpassing the 1B parameter threshold, using currently available toolkits, training these models in a HWA manner has become increasingly challenging. In this paper, we present AIHWKIT-Lightning, a fast and scalable toolkit for applying HWA training to large neural networks. Our contributions can be summarized as follows:

- We show AIHWKIT-Lightning outperforms existing toolkits in terms of speed and memory consumption across several benchmarks, including isolated layer performance and end-to-end training time;
- With the help of dedicated GPU kernels, we demonstrate a training speedup of up to $3.7\times$ over existing toolkits when the layers are split into smaller computational tiles;
- Finally, we show that by scaling HWA training up to more than 10B tokens, we achieve iso-accuracy on the GLUE benchmark using the RoBERTa model.

AIHWKIT-Lightning is publicly available at `https://github.com/IBM/aihwkit-lightning`.

## 2 Related Work

There exist many AIMC simulation frameworks/toolkits [13–17], each one with a different focus on functionalities such as on-chip inference simulation, HWA training, or simulated on-chip training. As outlined in [14], the libraries currently supporting HWA training are the AIHWKIT [13] and NeuroSIM [15]. The AIHWKIT is amongst the most popular toolkits for simulating inference and training of deep neural networks for AIMC hardware. It is primarily written in Python, CUDA, and C++, and builds upon the PyTorch framework. NeuroSIM is a framework that mostly focuses on performance estimation. Additionally, it supports the simulation of non-ideal device- and circuit-characteristics during inference and training, and hence, can be used to predict accuracy [15]. The HWA training capability of NeuroSIM is limited and does not offer as much configurability as AIHWKIT.

## 3 Background and Motivation

Recent publications of large scale integrated AIMC chips, using Phase Change Memory (PCM) [10, 18], Resisitive RAM (ReRAM) [12, 19, 20], or Flash [21], have demonstrated the viability of AIMC technology. Because of the cost and time involved in designing and fabricating chips, current systems often have a weight capacity of less than 50M weights. However, using simulations, the deployment of larger networks can be investigated. Recent works investigate the robustness of models such as BERT and RoBERTa [9, 22]. Still, training and evaluation of Large Language Models (LLMs) with more than 1B parameters has remained unexplored, mostly due to the computational demands and complexity of these networks.

Even with many GPUs working in parallel, training LLMs on billions of tokens is extremely compute intensive. Although current toolkits, such as NeuroSIM or AIHWKIT, support a wide array of features, HWA training of LLMs is, as we show in this paper, too inefficient for large-scale training. In the following section, we present AIHWKIT-Lightning, a complementary toolkit to AIHWKIT, with a refined focus on efficient and scalable HWA training.

## 4 AIHWKIT-Lightning

AIHWKIT-Lightning is designed to complement AIHWKIT by prioritizing ease-of-use and to accelerate HWA training. By using AIHWKIT-Lightning, users can train their models in a scalable fashion. For further evaluation, users can convert their model to AIHWKIT and use its rich inference simulation infrastructure. To make it easy for users to profit from the improved performance of AIHWKIT-Lightning, we structured the codebase similar to that of AIHWKIT. As a result, most training scripts using the AIHWKIT can be converted to AIHWKIT-Lightning by changing only the import statements from `from aihwkit.X import Y` to `from aihwkit_lightning.X import Y`.

By eliminating support for on-chip training and inference with different hardware models, our code is leaner, and the number of features available in the Resistive Processing Unit (RPU) configuration

– which is used to configure hardware and training related parameters – is drastically reduced. This simplifies the user-experience and reduces the number of pitfalls related to choosing the wrong setting. In the next section, we explain the HWA training features of AIHWKIT-Lightning in more detail.

## HW-Aware Training Features

The elementary layer of AIHWKIT-Lightning is the `AnalogLinear` layer, implementing the simple MVM operation $\mathbf{y} = \mathbf{x} \cdot \mathbf{W}^T$. In the following, we use $\mathbf{W}_{:,i}$ to refer to the $i-$th column of $\mathbf{W}$. Using this notation, the $i-$th output $y_i$ can be written as $y_i = \mathbf{x}^T \mathbf{W}_{:,i}$. To understand how the implementations of most HWA training features in AIHWKIT-Lightning differ from the ones in the AIHWKIT, it is important to also introduce the notion of normalized weights and inputs. In the AIHWKIT, the linear operation is performed using normalized inputs and weights $\mathbf{y}_{\text{norm}} = \mathbf{x}_{\text{norm}} \cdot \mathbf{W}_{\text{norm}}^T$, where

$$\mathbf{x}_{\text{norm}} = \mathbf{x}/\beta^{\text{inp. quant}}$$

$$\mathbf{W}_{:,i \text{ norm}} = \begin{cases} \mathbf{W}_{:,i}/\boldsymbol{\beta}_i^W & \text{if } \texttt{WeightRemapType}.\texttt{CHANNELWISE\_SYMMETRIC} \\ \mathbf{W}_{:,i}/\beta^W & \text{if } \texttt{WeightRemapType}.\texttt{LAYERWISE\_SYMMETRIC} \end{cases}.$$

Here, $\beta^{\text{inp. quant}}$ is a scalar used for normalizing the input to a configurable input bound and $\beta^W$ is a scalar or vector used for normalizing the weights per tensor or column, respectively. Because all of the necessary HWA training features can be realized without explicitly normalizing the weights and inputs, these conversions have been omitted in AIHWKIT-Lightning, resulting in a significant speedup of the `AnalogLinear` layer.

**Clipping.** When a HWA trained model is deployed on analog hardware using NVM devices, weights are mapped to conductance values and programmed into the NVM devices of the crossbar. Ensuring a tight weight distribution during training generally reduces the amount of programming noise and makes the mapping easier. During training, every weight tensor associated with an `AnalogLayer` is updated after every step of the optimizer, as described in Eq. 1.

$$\mathbf{W}^*_{:,i} \leftarrow \texttt{clamp}(\mathbf{W}_{:,i}, -\zeta_{\mathbf{i}}, \zeta_{\mathbf{i}}) \tag{1}$$

$$\zeta_{\mathbf{i}} = \begin{cases} \alpha \cdot \texttt{std}(\mathbf{W}_{:,i}) & \text{if } \texttt{WeightClipType}.\texttt{LAYER\_GAUSSIAN\_PER\_CHANNEL} \\ \alpha \cdot \texttt{std}(\mathbf{W}) & \text{if } \texttt{WeightClipType}.\texttt{LAYER\_GAUSSIAN} \end{cases}$$

Here, $\alpha$ is a tuneable hyperparameter that is typically set between $2.0$ and $3.5$. In AIHWKIT, clipping is implemented in a similar fashion, with the key difference that only the `WeightClipType`.`LAYER_GAUSSIAN` option exists. Per column clipping can be realized by normalizing the weights per column using the `WeightRemapType`.

**Analog Optimizer.** Similar to the AIHWKIT, any optimizer can be cast to an analog optimizer:

```
AnalogOptimizer(AdamW, model.analog_layers(), model.parameters(), lr=lr)
```

Unlike in the AIHWKIT, we also pass an iterator over the analog layers to the optimizer. This allows us to place a `step_post_hook` on the optimizer which iterates through the analog layers and applies clipping to the weights after every parameter update. This way, the `AnalogContext` parameter, required in the AIHWKIT, can be omitted.

**Noise Injection.** Besides guaranteeing a tight weight distribution, noise injection is also critical for achieving high accuracy when deploying neural networks on AIMC-based hardware [8]. AIHWKIT-Lightning supports two main sources of noise: weight and output noise. Weight noise is injected only during the forward pass. When backpropagating the gradients to the previous layer, noise-free weights are used. Weight noise is applied relative to the per column or per-tensor absolute maximum of the weight (see Eq. 2). The magnitude of the noise can be controlled with a scalar $\gamma$. Typical values of $\gamma$ lie between $0.02$ and $0.1$.

$$\mathbf{W}^{\text{noisy}}{}_{:,i} \leftarrow \mathbf{W}_{:,i} + \eta_{\mathbf{i}} \tag{2}$$

$$\eta_{\mathbf{i}} = \begin{cases} \gamma_{\text{weight}} \cdot \texttt{max}(\texttt{abs}(\mathbf{W}_{:,i})) \cdot \tau & \text{if } \texttt{WeightModifierType.ADD\_NORMAL\_PER\_CHANNEL} \\ \gamma_{\text{weight}} \cdot \texttt{max}(\texttt{abs}(\mathbf{W})) \cdot \tau & \text{if } \texttt{WeightModifierType.ADD\_NORMAL} \\ \text{where } \tau \sim \mathcal{N}(\mathbf{0}, \mathbf{I}) \end{cases}$$

Noise injected into the outputs can be modeled according to Eq. 3, where $\gamma_{\text{out}}$ is again a scalar controlling the relative amount of output noise injected. Note that the output noise is also relative to the input range, $\beta^{\text{inp. quant}}$, for a specific layer.

$$\mathbf{y}^{\text{noisy}}{}_{:,i} \leftarrow \mathbf{y}_{:,i} + \kappa_{\mathbf{i}} \tag{3}$$

$$\kappa_{\mathbf{i}} = \begin{cases} \gamma_{\text{out}} \cdot \beta^{\text{inp. quant}} \cdot \texttt{max}(\texttt{abs}(\mathbf{W}_{:,i})) \cdot \tau & \text{if } \texttt{forward.out\_noise.out\_noise\_per\_channel} \\ \gamma_{\text{out}} \cdot \beta^{\text{inp. quant}} \cdot \texttt{max}(\texttt{abs}(\mathbf{W})) \cdot \tau & \text{else} \\ \text{where } \tau \sim \mathcal{N}(\mathbf{0}, \mathbf{I}) \end{cases}$$

In contrast to AIHWKIT, noise samples are scaled relative to the inputs and weights, instead of explicitly normalizing the inputs and weights. As a result, AIHWKIT-Lightning requires the user to explicitly choose between the per column and per tensor mode, while this choice is made implicitly in the AIHWKIT by setting the `WeightRemapType`.

**Weight Quantization.** Although most AIMC architectures do not suffer from conventional weight quantization noise, it can still be helpful to represent weights using only a few bits. As shown in Eq. 4, symmetric quantization can be performed again per column or per tensor by choosing either the `WeightModifierType.DISCRETIZE_PER_CHANNEL` or `WeightModifierType.DISCRETIZE` option in the RPU configuration, respectively.

$$\mathbf{W}^{\text{quant}}{}_{:,i} \leftarrow \frac{\beta_i^{\text{weight quant}}}{2^{\text{weight bits}-1} - 1} \cdot \lfloor (\mathbf{W}_{:,i}) \cdot \frac{2^{\text{weight bits}-1} - 1}{\beta_i^{\text{weight quant}}} \rceil \tag{4}$$

$$\beta_i^{\text{weight quant}} = \begin{cases} \texttt{max}(\texttt{abs}(\mathbf{W}_{:,i})) & \text{if } \texttt{WeightModifierType.DISCRETIZE\_PER\_CHANNEL} \\ \texttt{max}(\texttt{abs}(\mathbf{W})) & \text{if } \texttt{WeightModifierType.DISCRETIZE} \end{cases}$$

In order to obtain quantized weights that are still robust to noise, one can also combine weight quantization with noise injection (see Eq. 2) by choosing the modes `WeightModifierType.DISCRETIZE_ADD_NORMAL{_PER_CHANNEL}`.

**Input Quantization.** In most AIMC-based heterogeneous architectures, Digital to Analog Converters (DACs) convert quantized inputs to voltage pulses that are then applied to the crossbar. AIHWKIT-Lightning supports symmetric static input quantization using learnable input ranges. Here static means that, at inference time, the input ranges are fixed and can not be adapted per input, as this would be too computationally expensive. Input quantization is performed according to Eq. 5.

$$\mathbf{x}^{\text{quant}} \leftarrow \frac{\beta^{\text{inp. quant}}}{2^{\text{input bits}-1} - 1} \cdot \lfloor \texttt{clamp}(\mathbf{x}, -\beta^{\text{inp. quant}}, \beta^{\text{inp. quant}}) \cdot \frac{2^{\text{input bits}-1} - 1}{\beta^{\text{inp. quant}}} \rceil \tag{5}$$

Similar to the AIHWKIT, we define a custom gradient for the input ranges. Given a scalar loss function $\mathcal{L}$ and the derivative w.r.t. the quantized input, $\nabla_{\mathbf{x}^{\text{quant}}}\mathcal{L}$, we can write the scalar derivative of $\mathcal{L}$ w.r.t. the input range $\beta^{\text{inp. quant}}$ as

$$\frac{\partial \mathcal{L}}{\partial \beta^{\text{inp. quant}}} = \beta^{\text{inp. quant}} \cdot \left( \sum (\nabla_{\mathbf{x}}\mathcal{L})_{\mathbf{x} \geq \beta^{\text{inp. quant}}} - \sum (\nabla_{\mathbf{x}}\mathcal{L})_{\mathbf{x} \leq -\beta^{\text{inp. quant}}} + \tau_{\text{decay}} \mathbb{1}\{p_{\text{not clip}} \geq T_{\text{not clip}}\} \right). \tag{6}$$

Where $\tau_{\text{decay}}$ is a decay hyperparameter that tightens the input range when the percentage of elements that did not get clipped, $p_{\text{not clip}}$, exceeds a user-definable threshold, $T_{\text{not clip}}$, which is typically set to 95%. The subscripts $\mathbf{x} \geq \beta^{\text{inp. quant}}$ and $\mathbf{x} \leq -\beta^{\text{inp. quant}}$ indicate positions where the input was above or below the input range, respectively, and the sum is taken over all the elements that fulfil this condition.

**Output Quantization.** When voltages are applied to the wordlines of the crossbar, current flows through the NVM devices and accumulates along the bitlines. The resulting current then flows into

Analog to Digital Converters (ADCs), where it is integrated for a certain period of time. We assume ADCs have a fixed current range, and hence, can be modelled using static quantization with fixed output bounds shared across all tiles [23], as shown in Eq. 7.

$$\mathbf{y}_i^{\text{quant}} \leftarrow \texttt{clamp}\left(\frac{\beta_i^{\text{adc quant}}}{2^{\text{adc bits}-1}-1} \cdot \left\lfloor \mathbf{y}_i \cdot \frac{2^{\text{adc bits}-1}-1}{\beta_i^{\text{adc quant}}} \right\rceil, -\beta_i^{\text{adc quant}}, \beta_i^{\text{adc quant}}\right) \tag{7}$$

$$\beta_i^{\text{adc quant}} = \begin{cases} \lambda_{\text{adc}} \cdot \beta^{\text{inp. quant}} \cdot \texttt{max}(\texttt{abs}(\mathbf{W}_{:,i})) & \text{if } \texttt{WeightClipType}.\texttt{CHANNELWISE\_SYMMETRIC} \\ \lambda_{\text{adc}} \cdot \beta^{\text{inp. quant}} \cdot \texttt{max}(\texttt{abs}(\mathbf{W})) & \text{if } \texttt{WeightClipType}.\texttt{LAYERWISE\_SYMMETRIC} \end{cases}$$

**Tiling.** The tiles in AIMC-based heterogeneous architectures are often smaller than most neural network layers. Therefore, MVMs with larger layers are executed across different tiles, and the partial results are accumulated using digital processing units. Training models with weight matrices tiled along the input dimension allows for more fine-grained input ranges and reduces the negative impact of weight outliers.

In the AIHWKIT, tiles are implemented as separate objects that, together, make up one analog module. When evaluating the module, separate MVM calls are invoked on the tiles and the results are aggregated.

AIHWKIT-Lightning leverages Triton [24] to hide the tiled computation of the analog linear layer in the blockwise nature of the matrix-matrix multiplication kernel. For an illustration, see Appendix Fig. 2. Using this approach, memory movement between DRAM and SRAM is drastically reduced, resulting in a significant decrease in latency.
The Triton mode can be activated by setting the environment variable `AIHWKIT_USE_TRITON`.

# 5 Results

All benchmarks presented in the following sections were obtained following best practices for benchmarking in PyTorch and Triton. For all benchmarks, we used an NVIDIA V100 GPU with 32GB of VRAM.

We observed that HWA training using NeuroSIM is roughly two times slower than AIHWKIT. This is in line with previous work comparing the two frameworks [25]. We will therefore limit most of our comparisons in this section to AIHWKIT, but include a layer-level comparison in Fig. 1.

**Benchmarking Single Layers.** First, we compare the latency of the `AnalogLinear` layer between AIHWKIT and AIHWKIT-Lightning for different HWA training configurations. More precisely, we measure the time it takes to do one training step, which includes one forward pass, one backward pass, and one optimizer step. We report the average latency over squared matrix dimensions ranging from 128 to 2'048 and a batch size of 512. For every HWA training configuration, we report the latency normalized w.r.t. the overall maximum latency. As a reference, we also show the time it takes to perform the operations using a standard PyTorch linear layer (see the dotted line in Fig. 1a). We show that AIHWKIT-Lightning consistently outperforms AIHWKIT and NeuroSIM for a variety of different HWA training settings.

**Tiled Layers.** Next, we compare the performance between AIHWKIT, AIHWKIT-Lightning, and the Triton mode of AIHWKIT-Lightning (denoted "Triton") for the same sequence of operations as before. This time, we leave the HWA training setting fixed (we use clipping, weight noise, and input range learning) and vary the square matrix dimensions between 1'024 and 4'608. We again use a batch size of 512. Fig. 1 compares the normalized latency between AIHWKIT, AIHWKIT-Lightning, and AIHWKIT-Lightning using the Triton mode. For the non-tiled MVM, AIHWKIT-Lightning consistently outperforms AIHWKIT and the Triton mode is the fastest up to a layer size of 2'048. When the layer is tiled across the input dimension (here a maximum input size of 512), we see that the Triton kernel significantly outperforms AIHWKIT and AIHWKIT-Lightning with the Triton mode turned off.

**Benchmarking End-to-End Training.** Although benchmarking individual components, such as individual layers, is more indicative of the true speedup AIHWKIT-Lightning yields over AIHKWIT, it is also important to consider a real-world setting where users would optimize training hyperparameters, such as the batch size or the number of gradient accumulation steps.

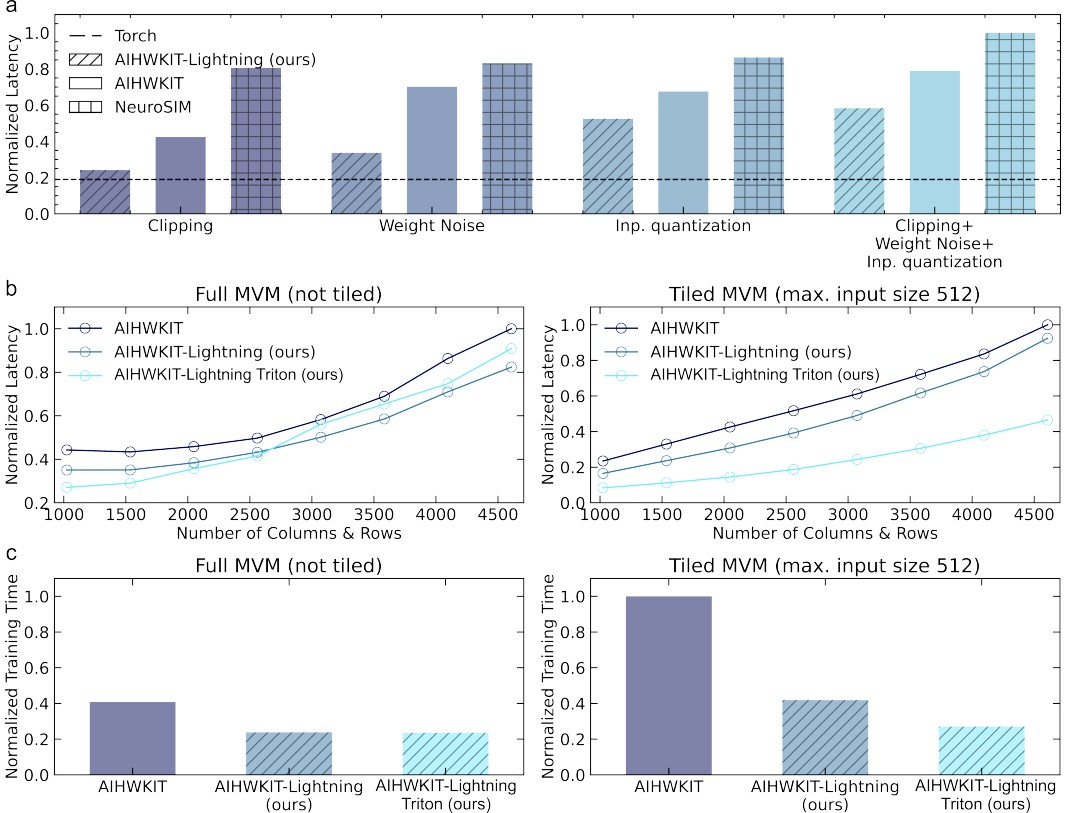

Figure 1: **a.** Single layer comparison between AIHWKIT-Lightning, AIHWKIT, and NeuroSIM. **b.** Non-tiled vs. tiled MVM comparison. **c.** End-to-end training time comparison.

We measured the time to pre-train the T5-large model with 770M parameters on WikiText-103. For this benchmark we also used a combination of clipping, weight noise injection and input range learning. Note that because of the reduced memory footprint of AIHWKIT-Lightning, we were able to increase the batch size compared to AIHWKIT from one to two, giving us an additional training speedup. For a more thorough investigation of the memory consumption, see Appendix B. Fig. 1c shows the training time comparison between AIHWKIT, AIHWKIT-Lightning, and AIHWKIT-Lightning using the Triton mode for non-tiled and tiled `AnalogLinear` layers. The training times are normalized against the highest training time for tiled `AnalogLinear` layers. We find that for non-tiled layers, AIHWKIT-Lightning and AIHWKIT-Lightning with the Triton mode outperform AIHWKIT by $1.72\times$. When using the tiled `AnalogLinear` layer, the Triton mode yields a speedup of $3.7\times$ over AIHWKIT, training $1.5\times$ faster than the AIHWKIT implementation using non-tiled `AnalogLinear` layers. When pre-training T5-large on WikiText-103 using 8 V100 GPUs in parallel, AIHWKIT-Lightning takes 34.1 hours, while AIHWKIT takes 58.8 hours, which is a difference of more than a day.

Using AIHWKIT-Lightning, we were also able to train Phi-3-Mini [26] (3.8B parameters) on 1B tokens in under 6 hours using 96 V100 GPUs. Because of the increased memory fragmentation, training Phi-3-Mini using AIHWKIT resulted in an out-of-memory exception.

**Accuracy Study.** Using this more scalable approach of HWA training, we can perform HWA training on more data. We demonstrate this by HWA training RoBERTa [27] on 11B tokens using the conventional masked language modeling objective. We performed distributed data parallel training on 8 NVIDIA V100 GPUs for 35 hours. After repeating part of the pre-training phase in a HWA manner, we further finetuned the model for each GLUE task. When HWA training is introduced **only** during the finetuning stage, average performance is off by 2.71% compared to the FP16 model. By doing HWA training on more data, we can close this gap to 0.74%. For further information, see Appendix B.

# 6 Conclusion

We presented AIHWKIT-Lightning, a toolkit for scalable HWA training of large neural networks. By using dedicated GPU kernels for HWA training, we demonstrated a speedup of up to $3.7\times$ over the current state-of-the-art toolkit. While we have demonstrated training of LLMs with up to 3.8B parameters, training larger LLMs might require further optimizations such as more optimized GPU kernels or faster alternatives to some HWA training features. We believe that AIHWKIT-Lightning enables researchers to perform HWA training at scale, and hope that this contribution opens up new insights into the robustness of larger models, such as LLMs.

## Acknowledgments and Disclosure of Funding

This work was supported in part by the IBM Research AI Hardware Center; in part by the European Union's Horizon Europe Research and Innovation Programme under Grant 101046878; and in part by the Swiss State Secretariat for Education, Research and Innovation (SERI) under Contract 22.00029.

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

# A  Memory Consumption and Fragmentation

The AIHWKIT requires memory allocation of some additional tensors such as scales which normalize weights. When converting a model to an analog model, more allocations occur with a smaller mean allocation size. For example, converting RoBERTa to an analog model in AIHWKIT triggers 576 memory allocations with an average allocation size of 1.13MB. In comparison, AIHWKIT-Lightning only triggers 502 allocations with an average size of 1.3MB. One can see that, although both frameworks roughly allocate the same amount of memory ($\sim$650MB), AIHWKIT-Lightning leads to less memory fragmentation. In a tiled setting, this effect is more pronounced, as the AIHWKIT creates a separate object for every tile. For a maximum input size of 512, the AIHWKIT now triggers 1308 allocations with an average size of only 0.5MB. The result of increased fragmentation is that, despite sufficient memory capacity, larger models can not be loaded into using the AIHWKIT.

During the forward and backward pass, the AIHWKIT consumes approximately $1.2\times$ more memory compared to AIHWKIT-Lightning, as depictred in Fig. 1. To perform this comparison, we used a linear layer of size $4'096 \times 4'096$ and an input with a batch size of $4'096$.

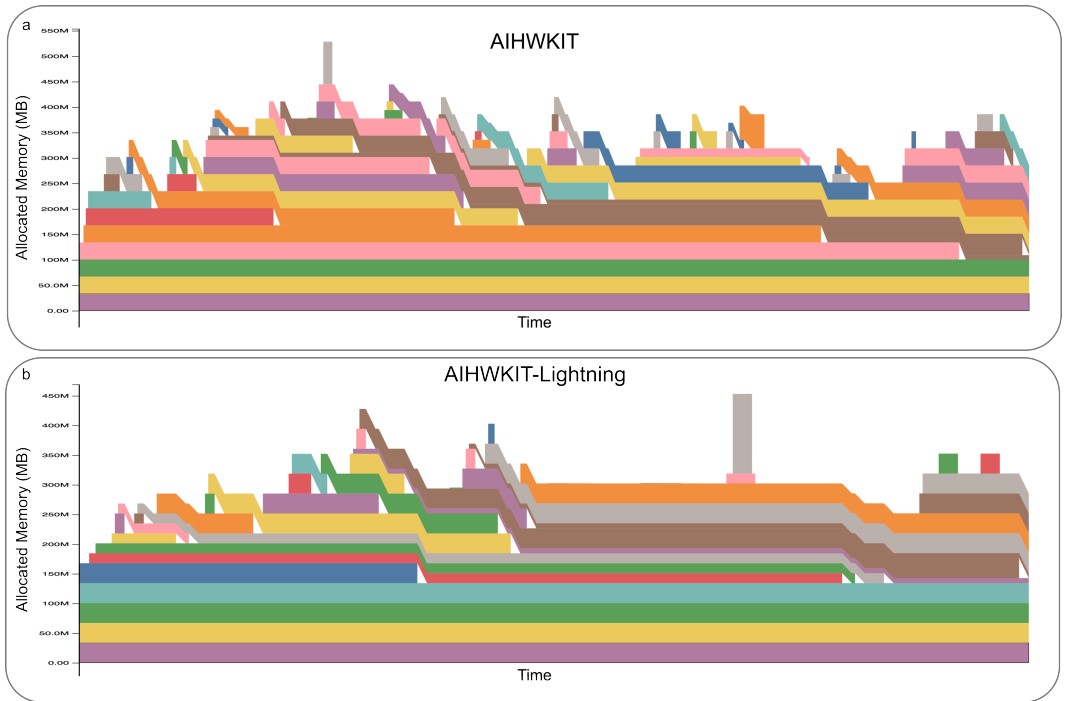

Appendix Figure 1: Comparison of memory consumption during forward and backward pass. Every colored bar shows the lifetime of one allocated tensor. **a.** Trace for the AIHWKIT. **b.** Trace for AIHWKIT-Lightning.

# B  Accuracy Study

We use the following HWA training features. We clip the weights to $\alpha = 2.5$ standard deviations, quantize the inputs to 8 bits, and learn the input ranges, and we inject noise of magnitude $\gamma_{\text{weight}} = 0.023$ into the weights. The clipping and the noise injection is done per tensor, and not per column. During evaluation, we fix the input ranges for input quantization and simulate programming noise introduced when programming target conductances into the NVM devices of the crossbars. The noise model was obtained from the IBM HERMES Project Chip [10]. Appendix Table 1 shows the results on every GLUE task. Evaluations using the noise model were repeated five times.

Appendix Table 1: Comparison of model performance on the GLUE benchmark with different HWA training procedures.

| Model | RTE | QQP | QNLI | CoLA | MRPC | STSB | SST2 | MNLIm | Avg |
|---|---|---|---|---|---|---|---|---|---|
| **Evaluated in FP16** | | | | | | | | | |
| **Pre-trained model** | 78.34 | 91.72 | 92.81 | 62.19 | 89.46 | 90.92 | 94.72 | 87.82 | 85.98 |
| **Evaluated with hardware noise model and INT8 static input quantization** | | | | | | | | | |
| **HW-Aware** (only finetuning) | 73.61 ± 2.01 | 91.29 ± 0.05 | 92.12 ± 0.21 | 51.71 ± 2.01 | 87.01 ± 0.51 | 90.04 ± 0.15 | 93.89 ± 0.35 | 86.53 ± 0.16 | 83.27 |
| **HW-Aware** (pre-training + finetuning) | 74.87 ± 0.51 | 91.42 ± 0.42 | 92.41 ± 0.17 | 62.48 ± 1.03 | 88.33 ± 1.17 | 90.52 ± 0.18 | 94.70 ± 0.35 | 87.21 ± 0.07 | **85.24** |

# C  Tiling

The general matrix multiply algorithm can be implemented in a block-wise fashion using Triton (`https://triton-lang.org/main/index.html`). Here, the elementary operations do not operate on single elements, but rather on blocks of data. At the heart of the algorithm is a loop that iteratively loads the input and weight blocks along the hidden dimension, performs the block-wise dot product, and accumulates the result in a buffer, which is then written to the output tensor. In the case of a tiled `AnalogLinear` layer, input and weight blocks belong to different tiles. When iterating over the hidden input dimension, input blocks are loaded. After determining to which tile the block belongs to, input scaling and quantization using the tiles' input ranges are applied. The dot product with the corresponding weight block is then written in a per tile accumulator block. After the row-column-loop completes, every per tile accumulator contains the correct per tile MVM result. At that stage, ADC quantization and output noise can be applied and the per tile accumulators can be combined into the final block that is written to the output tensor. This procedure is illustrated in Fig 2.

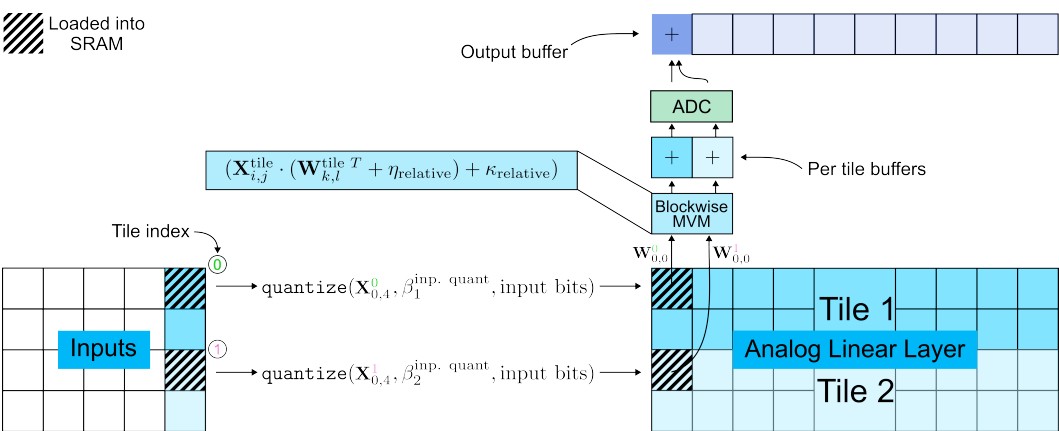

Appendix Figure 2: Illustration of the Triton kernel implementing the tiled analog linear layer.

