# OpenReview forum: "AIHWKIT-Lightning: A Scalable HW-Aware Training Toolkit for Analog In-Memory Computing"
_NeurIPS.cc/2024/Workshop/MLNCP — MLNCP Poster_

### Official Review · Reviewer_n93o · 2024-10-04
**Improvement in terms of scalability on AIHWKIT**

**Rating:** 7
**Confidence:** 2

**Review:**

This toolkit addresses the limitations of existing frameworks by improving training speed (up to 3.7x faster), reducing memory consumption, and achieving near-iso-accuracy on benchmarks. The paper highlights the reduction of memory fragmentation and overall memory consumption, which allows larger models with billions of parameters to be trained more efficiently. Furthermore the framework is presented as an easy-to-integrate solution for existing AIHWKIT users.

Authors can improve upon these questions:

Although the paper mentions accuracy drops of 2.71% when HWA training is introduced during fine-tuning, it only briefly discusses the methods used to mitigate this drop.

While the toolkit is benchmarked extensively in a GPU environment, the paper does not yet demonstrate its performance on physical AIMC hardware. Testing on real hardware would provide further validation of the toolkit's effectiveness

---

### Decision · Program_Chairs · 2024-10-10

Accept (Poster)